# Synthesis of Novel Conjugated Linoleic Acid (CLA)-Coated Superparamagnetic Iron Oxide Nanoparticles (SPIONs) for the Delivery of Paclitaxel with Enhanced In Vitro Anti-Proliferative Activity on A549 Lung Cancer Cells

**DOI:** 10.3390/pharmaceutics14040829

**Published:** 2022-04-11

**Authors:** Lindokuhle M. Ngema, Samson A. Adeyemi, Thashree Marimuthu, Philemon Ubanako, Daniel Wamwangi, Yahya E. Choonara

**Affiliations:** 1Wits Advanced Drug Delivery Platform Research Unit, Department of Pharmacy and Pharmacology, School of Therapeutic Sciences, Faculty of Health Sciences, University of the Witwatersrand, 7 York Road, Parktown, Johannesburg 2193, South Africa; lindokuhle.ngema@students.wits.ac.za (L.M.N.); samson.adeyemi@wits.ac.za (S.A.A.); thashree.marimuthu@wits.ac.za (T.M.); philemon.ubanako@wits.ac.za (P.U.); 2School of Physics, Materials Physics Research Institute, University of the Witwatersrand, Private Bag 3, WITS, Johannesburg 2050, South Africa; daniel.wamwangi@wits.ac.za

**Keywords:** superparamagnetic iron oxide nanoparticles, non-small-cell lung carcinoma, A549 cell line, conjugated linoleic acid, paclitaxel, nanomedicine

## Abstract

The application of Superparamagnetic Iron Oxide Nanoparticles (SPIONs) as a nanomedicine for Non-Small Cell Lung Carcinoma (NSCLC) can provide effective delivery of anticancer drugs with minimal side-effects. SPIONs have the flexibility to be modified to achieve enhanced oading of hydrophobic anticancer drugs such as paclitaxel (PTX). The purpose of this study was to synthesize novel *trans*-10, *cis*-12 conjugated linoleic acid (CLA)-coated SPIONs loaded with PTX to enhance the anti-proliferative activity of PTX. CLA-coated PTX-SPIONs with a particle size and zeta potential of 96.5 ± 0.6 nm and −27.3 ± 1.9 mV, respectively, were synthesized. The superparamagnetism of the CLA-coated PTX-SPIONs was confirmed, with saturation magnetization of 60 emu/g and 29 Oe coercivity. CLA-coated PTX-SPIONs had a drug loading efficiency of 98.5% and demonstrated sustained site-specific in vitro release of PTX over 24 h (i.e., 94% at pH 6.8 mimicking the tumor microenvironment). Enhanced anti-proliferative activity was also observed with the CLA-coated PTX-SPIONs against a lung adenocarcinoma (A549) cell line after 72 h, with a recorded cell viability of 17.1%. The CLA-coated PTX-SPIONs demonstrated enhanced suppression of A549 cell proliferation compared to pristine PTX, thus suggesting potential application of the nanomedicine as an effective site-specific delivery system for enhanced therapeutic activity in NSCLC therapy.

## 1. Introduction

Non-Small Cell Lung Carcinoma (NSCLC) is the most prevalent type of lung cancer affecting millions of people worldwide, with almost 85% of diagnosed lung cancer cases being NSCLC [1]. The treatment of NSCLC continues to be a daunting challenge, with combinational chemotherapy as first-line treatment reportedly at its therapeutic plateau [2]. Intolerable toxicity due to high doses and non-specificity are key hindrances in the effective use of combinational chemotherapy [3,4,5,6]. Paclitaxel (PTX) is a first-line anticancer drug used in platinum-based combination chemotherapy for advanced NSCLC [7,8]. PTX shows adequate efficacy at the onset of treatment, however, subsequent side effects (i.e., neutropenia, myalgia, allergic reactions) and resistance are major setbacks for effective NSCLC treatment [9]. Consequently, numerous studies have proposed the use of targeted nano-enabled drug delivery systems, including nanoparticles and nanoemulsions, as an approach to enhance the efficacy of PTX in NSCLC therapy [1,10,11,12,13,14].

For example, the use of Superparamagnetic Iron Oxide Nanoparticles (SPIONs) as anticancer drug delivery vectors has shown enormous success over the years [15,16,17,18]. SPIONs offer desirable features as drug transporters for NSCLC therapy, such as high drug-loading capacity, small size (<100 nm) to allow penetration into lung tumor, and a biocompatible outer-shell that can be functionalized with targeting ligands to achieve active targeted drug delivery [19,20,21]. Studies have shown that anticancer drugs loaded into SPIONs present with high bioavailability [19,22,23], and coating prevents particle aggregation and extends the systemic circulation time of SPIONs [24,25]. In addition, coating of SPIONs with naturally derived anticancer compounds, such as conjugated linoleic acid, could render them as excellent nanovectors for targeted NSCLC therapy.

Conjugated Linoleic Acid (CLA) has been previously explored to have potential anticancer activity and comprises of ~28 isomers [26,27]. CLA has been shown to suppress proliferation in breast [28], colon [29], stomach [30], colorectal [31], and prostate cancer cells [32]. The *trans*-10, *cis*-12 CLA isomer (10E, 12Z CLA) is one of two biologically active isomers of CLA. The 10E, 12Z CLA reportedly acts on adipocytes where it limits lipid uptake by blocking stearoyl-CoA desaturase and lipoprotein lipase activity and affects lipid metabolism in cancer cells, which subsequently halts cell growth and metastasis [33,34]. Tumor cells require high energy supply to proliferate and metastasize, hence affecting lipid metabolism pathways, which halts their growth and metastasis [35,36]. In addition, the hydrophobic nature of CLA is suitable for partitioning hydrophobic drugs such as PTX, thus allowing higher drug loading capacity and encapsulation into nanostructures.

Therefore, in this study, we describe the novel inclusion of CLA as a surface directing agent for efficient drug loading on SPIONS for potential targeted therapy in NSCLC. The combination of PTX and SPIONs with a natural fatty acid such as CLA (with potential anticancer activity) could yield a more efficacious nanomedicine for NSCLC chemotherapy with enhanced anti-proliferative activity and biocompatibility. To investigate the effect of CLA functionalization of SPIONS, a nanosystem comprising 10E, 12Z CLA surface-coated SPIONs with self-assembly of PTX (termed CLA-coated PTX-SPIONs) was synthesized and characterized for all pertinent physiochemical, pharmaceutical (including anti-proliferative activity on an A549 lung cancer cell line), and superparamagnetic properties. The results of this work will provide new insights into the design of nanomedicines for NSCLC.

## 2. Materials and Methods

### 2.1. Materials

Metal precursors comprising of iron (III) chloride hexahydrate (FeCl_3_∙6H_2_O) and iron (II) chloride tetrahydrate (FeCl_2_∙4H_2_O), key reagents (sodium hydroxide (NaOH) pellets and conjugated (10E, 12Z)-linoleic acid solution in ethanol), paclitaxel (PTX), and consumables inclusive of cell culture assays (tween 80, phosphate buffered saline tablets, magnetite standard, SnakeSkin™ dialysis membrane, 3500 MWCO, Foetal Bovine Serum (FBS), penicillin-streptomycin antibiotics, dimethyl sulfoxide (DMSO), and acid-isopropanol solubilizing agent) were procured from Sigma-Aldrich (St. Louis, MO, USA). Solvents (methanol and ethanol) classified as analytical grade were used without additional purification.

### 2.2. Fabrication of the CLA-Coated, PXT-Loaded SPIONs (CLA-PTX-SPIONs)

#### 2.2.1. Synthesis of the SPIONs

To ensure high yield, a co-precipitation method was adopted for the preparation of SPIONs [37]. Briefly, an aqueous solution of FeCl_2_∙4H_2_O (0.3 mol) and FeCl_3_∙6H_2_O (0.6 mol) was prepared in 60 mL N_2_-purged deionized water to attain 1:2 M ratio of Fe^2+^ and Fe^3+^. Thereafter, 0.4 M NaOH (pH = 13) was added drop-wise into the solution under continuous stirring (300 rpm) at 75 °C until dark precipitates of SPIONs formed. This was executed under N_2_ gas to eliminate oxygen and control the size of SPIONs produced. The formed SPIONs were centrifuged at 5000 rpm (Eppendorf Centrifuge 5804, Hamburg, Germany) for 30 min at room temperature (25 °C) to eliminate any unreacted material. The precipitate (SPIONs) was washed twice with N_2_-purged deionized water and separated by magnetic decantation and air dried.

#### 2.2.2. Coating of the SPIONs with *trans*-10 *cis*-12 (10E, 12Z) Conjugated Linoleic Acid

Coating of the SPIONs was carried out in a single-step reaction. Forty microliters from a 100 mg/mL 10E, 12Z CLA solution in ethanol was added drop-wise into a 1.5 mL suspension of SPIONs (20 mg SPIONs in 1.5 mL ethanol) in a sonication bath (Sientech Ultrasonic cleaner, Labotec, Midrand, South Africa) kept at 80 °C and at low frequency for 30 min. A magnet was used to precipitate the coated SPIONs. The supernatant was withdrawn, and the precipitate washed twice with deionized water and ethanol, respectively. The CLA-coated SPIONs (10E, 12Z CLA-SPIONs) were vacuum dried at 80 °C for 24 h [38,39]. The quantity of CLA on SPIONs was then determined by thermogravimetric analysis (TGA).

#### 2.2.3. Self-Assembled Loading of PTX into 10E, 12Z CLA-SPIONs

In order to exploit SPIONs as an anticancer drug carrier, paclitaxel (PTX) was self-assembled on the hydrophobic tails of CLA coated on SPIONs surface. To achieve this, a superlative method by Dilnawaz et al. [40] was employed, with modifications. Accordingly, 10% (*w*/*w*) PTX was used for loading into 100% (*w*/*w*) CLA-coated SPIONs. A total of 15 mg CLA-coated SPIONs was dispersed in 1.5 mL methanol and sonicated for 2 min. PTX (1.5 mg) was dissolved in 70% methanol (150 µL) and added drop-wise to the CLA-coated SPIONs suspension under continuous gradual stirring at 200 rpm (Magnetic Stirrer MSH10, Labcon, Johannesburg, South Africa) overnight to allow for the adsorption of PTX onto the hydrophobic CLA on the surface of the SPIONs. The particle suspension was centrifuged at 13,800 rpm (TC-MiniSpin Centrifuge, TopScien, Ningbo, China) for 10 min at 10 °C, with subsequent washing (three times distilled water) to separate free drug. The supernatant (with free drug) was collected for quantifying the PTX loading capacity and adsorption efficiency via UV spectrophotometry (Cary™ 50, Varian Inc., Palo Alto, CA, USA). The PTX-loaded CLA-coated SPIONs subsequently lyophilized (FreeZone-50C Benchtop Freeze Dryer, Labconco, Kansas City, MO, USA) to attain free-flowing powdered CLA-coated PTX-SPIONs.

### 2.3. Physicochemical Characterization of the Synthesized CLA-Coated PTX-SPIONs

#### 2.3.1. Analysis of Chemical Structure Integrity and Transitions

FTIR spectroscopy (Spectrum 100, PerkinElmer Inc., Waltham, MA, USA) was appropriately used to comparatively analyze vibrational transitions in the chemical structures of the individual and combined components of the CLA-coated PTX-SPIONs. Samples of pristine SPIONs, CLA, PTX, and CLA-coated PTX-SPIONs were analyzed at 120 psi pressure in a range of 4000–550 cm^−1^ over 20 scans to obtain relevant spectra.

#### 2.3.2. Determination of Particle Size, Zeta Potential, and Morphology

Dynamic light scattering and phase-analysis light scattering measurements to obtain the average hydrodynamic diameter and zeta potential, respectively, of the synthesized SPIONs was implemented on a ZetaSizer instrument (NanoZS, Malvern Panalytical, Malvern, UK). Dried samples (10 µg) of pristine SPIONs and CLA-coated PTX-SPIONs were suspended in 1 mL distilled water followed by sonication for 10 min (Sonics Vibra Cell, Newtown, Newtown, CT, USA). Samples were transferred into disposable cuvettes for hydrodynamic size and polydispersity index (PDI) analysis, thereafter into zeta potential cuvettes (DTS 1070) for surface charge analysis at 25 °C.

A sample of CLA-coated PTX-SPIONs was also analyzed for morphology using Scanning Electron Microscopy (SEM) (ZEISS Electron Microscopy, Carl Zeiss Microscopy Ltd., Cambridge, UK). An aliquot was carefully placed on an aluminum specimen stub and air dried before coating (×2) with gold palladium (AuPd). The sample was viewed and captured under a low (82.34 KX) and high (314.84 KX) magnification.

#### 2.3.3. Determination of CLA Content on CLA-Coated SPIONs

The amount of CLA coated onto SPIONs was determined using thermogravimetric analysis (TGA) (TGA 4000, PerkinElmer Inc., Waltham, MA, USA). Pristine SPIONs and CLA-coated SPIONs were analyzed (mass change as function of temperature over time) under set TGA conditions: 30–900 °C, continuous nitrogen flow, and heat rate of 10 °C/min.

#### 2.3.4. Analysis of the Crystallographic Structure of the CLA-Coated PTX-SPIONs

X-ray diffractometry (XRD) (RIGAKU MiniFlex, RIGAKU Inc., Tokyo, Japan) was employed to analyze the crystallinity of the SPIONs before and after modifications. Dried samples of pristine SPIONs and the CLA-coated PTX-SPIONs were mounted on alumnium sample holders and analyzed to obtain diffractograms. The following measurement conditions were set for the analysis: 2θ range of 5°–90°, scan rate at 10 deg/min, current and voltage were 15 mA and 40 kV, respectively.

#### 2.3.5. Iron Oxide Core Mapping of the SPIONs

Raman spectroscopy (SENTERRA II, BRUKER, Bremen, Germany) was employed because it is a precise method to characterize the magnetite nature of the iron oxide core of the SPIONs before and after the CLA-coating enhancement process [38,41]. Samples were prepared on glass slides and analyzed with a beam diameter of 1μm at a laser power of 0.04 mV at 25 °C. A magnetite standard (Sigma-Aldrich, Saint Louis, MO, USA) was used as reference.

#### 2.3.6. Analysis of the Magnetic Properties of the CLA-Coated PTX-SPIONs

The paramagnetic properties of the synthesized SPIONs and the CLA-coated PTX-SPIONs were analyzed using a vibrating sample magnetometer (VSM) (12T Physical Property Measurement System, PPMS^®^ DynaCool™, Quantum Design, San Diego, CA, USA), at room temperature. The PPMS^®^ DynaCool™ measurement temperature was fixed at 300 K with the magnetic field span of −8000 to 8000 Oe.

#### 2.3.7. Determination of PTX-Loading Capacity and Adsorption Efficiency within the CLA-Coated SPIONs

A standard calibration curve with serial concentrations (5–25 µg/mL) was prepared for PTX loading using UV spectrophotometry (Cary™ 50, Varian Inc., Palo Alto, CA, USA). The concentrations were prepared in a 70:30 methanol: water co-solvent system. The Adsorption Efficiency (%AE) and Drug (PTX) Loading Capacity (%DLC) were defined by quantification of free PTX in the supernatant following the loading procedure. Briefly, the PTX-loaded CLA-coated SPIONs were centrifuged (13,800 rpm, 10 min, 10 °C) with subsequent washing, thrice with distilled water, and the supernatant was withdrawn. The concentration and the amount of PTX in the supernatant were determined by recording the absorbance of the samples (*n* = 3) at 230 nm, and the %AE as well as %DLC were computed by application of Equations (1) and (2), respectively.
(1)$$\%\mathrm{AE}=\frac{D_{l}-D_{s}}{D_{l}}\times 100$$
(2)$$\%\mathrm{DLC}=\frac{D_{l}-D_{s}}{W_{f}}\times 100$$
where *D_l_* is the amount of drug loaded, *D_s_* is the amount of unloaded drug in supernatant, and *W_f_* is the total weight of formulation, in mg.

#### 2.3.8. Evaluation of In Vitro Release of PTX from the CLA-Coated SPIONs

The drug release profile was evaluated for the CLA-coated PTX-SPIONs at pH 6.8 and pH 7.4 for 24 h at 37 °C, mimicking tumor microenvironment (TME) and physiological pH, respectively [40,42]. Lyophilized CLA-coated PTX-SPIONs (3 mg) were dispersed in 2 mL of prepared 0.1 M PBS (pH: 6.8 and, 7.4) containing 0.1% *v*/*v* Tween. The samples were then loaded into a dialysis membrane (SnakeSkin™, 3500 MWCO) and submerged into the relevant drug release buffer (30 mL) of corresponding pH and incubated in an orbital shaking incubator (YIHDER LM-530, YIHDER Co., Ltd., Taipei, Taiwan). The dialysis membrane was used to facilitate only free PTX diffusion into the release medium for assay. Sampling (2 mL) was undertaken at different time intervals (*t* = 1, 2, 4, 8, 12, and 24 h) and the UV absorbance readings were set at 230 nm for PTX. The release medium was replenished with the same amount of drug-free volume sampled to maintain sink conditions.

### 2.4. In Vitro Evaluation of the Anti-Proliferative Activity of the CLA-Coated PTX-SPIONs for Potential Treatment of NSCLC Using a MTT Assay

The model lung adenocarcinoma cell line (A549) (Cellonex, Johannesburg, South Africa) was selected for the cell viability study. In a typical protocol, cells in Dulbecco’s Modified Eagle’s Medium (DMEM) were cultured in a T-25 culture flask and supplementation with Foetal Bovine Serum (FBS, 10% (*v*/*v*)) for and penicillin-streptomycin antibiotics 1% (*v*/*v*) was carried out prior to incubation at 37 °C and 5% CO_2_ saturation under humid conditions. Under microscopic visualization when a 90% confluence was reached, the cells were transferred and seeded in a 96-well plate, at a seeding density of 2.5 × 10^4^. To each well, a cell suspension of 90 µL was added followed by a 96-well plate incubation at 37 °C, 5% CO_2_ for 24 h to allow adequate cellular adherence. The experimental cells were then treated in triplicate (*n* = 3) with 10 µL of SPIONs, CLA, and CLA-coated PTX-SPIONs samples separately to produce treatment concentrations of 3.125, 6.25, 12.5, 25, 50, and 100 µg/mL in PBS (0.2% DMSO). Pristine PTX was used as a positive control at varying treatment concentrations of 0.156, 0.312, 0.625, 1.25, 2.5, 5, and 10 µg/mL. The cells were subjected to incubation for 72 h prior to the commencement of the cell viability assay.

The MTT assay was performed after 72 h using the MTT Cell Proliferation Kit I (Roche, Basel, Switzerland). MTT solution (10 µL; 5 mg/mL) was introduced into each well, followed by 4-h incubation of the plate (37 °C, 5% CO_2_, humid conditions). Thereafter, 100 µL of the solubilizing agent (acid-isopropanol; 0.04 N HCl in isopropanol) was added to dissolve formed formazan crystals, followed by overnight incubation at 37 °C and 5% CO_2_. Wells containing only solubilizing agent and growth medium were used as a blank. The absorbance was measured with a multimodal microplate reader (Victor X3, PerkinElmer, Waltham, MA, USA) at 570 nm with a reference wavelength of 620 nm. The absorbance measurements were computed (mean ± standard deviation) and used to calculate % cell viability via Equation (3).
(3)$$\%\;\mathrm{Cell}\;\mathrm{viability}=\frac{A_{test}-A_{blank}}{A_{control}-A_{blank}}\times 100$$
where *A_test_* is test absorbance, *A_blank_* is blank absorbance, and *A_control_* is control absorbance in nm.

The AAT Bioquest IC_50_ calculator (ATT Bioquest, Inc., Sunnyvale, CA, USA) was used to compute IC_50_ values [43].

### 2.5. Statistical Analysis

The results were analyzed using a two-tailed Student’s unpaired *t*-test on GraphPad prism version 9 (GraphPad Software, Inc., San Diego, CA, USA). Two groups were analyzed at a time at 95% confidence interval (*p <* 0.05). All data are expressed as mean ± standard deviation of analyses in triplicate (*n* = 3).

## 3. Results and Discussion

### 3.1. Assessment of the CLA-Coated PTX-SPIONs Synthesized in Terms of Particle Size and Morphology

Amongst several reported methods of synthesizing magnetic nanoparticles (MNPs) (including SPIONs), co-precipitation was selected as the most suitable as it is undertaken in an inert atmosphere with alkalinity and high temperature to avoid oxidation and to control particle size, distribution, and morphology [44,45]. In the present study, SPIONs were successfully synthesized as a black colored powder (characteristic of pure Fe_3_O_4_ nanoparticles). Essentially pure magnetite Fe_3_O_4_ exhibits a black color and once oxidized it transitions to a brownish/red color [46]. Performing the precipitation in an inert environment (N_2_) evidently prevented oxidation/reduction of the iron II/III salts and the deterioration of the SPIONs. Furthermore, coating of the synthesized SPIONs with fatty acids such as CLA (having a carboxylic group) can limit particle agglomeration and prevents oxidation [38].

In this study, CLA was used to coat the SPIONs mainly due to its previously reported anticancer properties and interestingly the pharmaceutical formulation benefits owing to the inherent carboxyl group, its hydrophobicity to partition poorly soluble drugs, and excellent crosslinking activity [38,47]. The successful coating of CLA onto the SPIONs surface occurred through chemisorption via the carboxyl group of CLA and the oxygen coordinated at the surface of the SPIONs. A similar mechanism was reported for SPIONs coated with oleic acid and glycerol monooleate, both containing carboxyl groups [38,39,40]. The hydrophobicity of CLA further allowed efficient loading of PTX (hydrophobic drug) onto the CLA-coated SPIONs via spontaneous hydrophobic interactions. PTX was adsorbed on the CLA hydrophobic ends surrounding the SPIONs core to form the CLA-coated PTX-SPIONs [48,49].

The average hydrodynamic particle size and zeta potential of pristine SPIONs was measured at 51.0 ± 1.3 nm and −24.3 ± 1.3 mV, respectively, while CLA-coated SPIONs exhibited average size of 62.7 ± 5.6 nm and zeta potential of −26.3 ± 0.5 mV. The CLA-coated PTX-SPIONs had a hydrodynamic size of 96.5 ± 0.6 nm and a zeta potential of −27.3 ± 1.9 mV (Figure 1a,b). The synthesized SPIONs were therefore in a desirable range for potential targeted PTX delivery in NSCLC. Essentially, previous studies have shown that conventional MNPs in a 10–100 nm range can penetrate lung tumors, while particles <10 nm are subject to renal clearance, and those >100 nm accumulate in the spleen or are removed by alveolar macrophages in the lungs [50,51]. The resultant particle size for SPIONs produced in this study is attributed to the co-precipitation method of synthesis and the processing parameters that were maintained as constant (i.e., 75 °C temperature and inert atmosphere). Processing factors such as pH, reaction temperature, and media are known to significantly influence the particle size [52,53].

Interestingly, the pristine SPIONs had a PDI value of 0.4 ± 0.1, while the CLA-coated PTX-SPIONs were recorded at 0.1 ± 0.02. This indicated that the CLA-coating reduced aggregation and imparted a more colloidal stability to the particles. These results, together with the zeta potential values recorded for the CLA-coated PTX-SPIONs, reveal that this synthesis strategy provides superior dispersion and stability in aqueous media, suggesting its suitability as nano-enabled drug delivery system. Modified SPIONs in a similar particle size range and zeta potential value have been previously reported by Goncalves et al. [54]. The overall morphology of the free-flowing powdered CLA-coated PTX-SPIONs (Figure 1c), as confirmed by SEM imagery (Figure 1d,e), further revealed that the CLA-coated PTX-SPIONs were spherical in shape and had a relatively rugged surface with a mean particle size of <100 nm. This corroborates with previous studies that reported on co-precipitation yielding spherical nanoparticles [37,55], which when employed in cancer nanomedicine studies, result in enhanced biodistribution and cellular uptake [56,57].

### 3.2. Assessment of the Chemical Stability and Functional Transformation of the CLA-Coated PTX-SPIONs

The FT-IR spectra of pristine SPIONs, CLA, and PTX as well as the drug-free and drug-loaded CLA-coated SPIONs are presented in Figure 2a–e. Two bands were observed on the spectrum of pristine (uncoated) SPIONs (Figure 2a) at around 3420 cm^−1^ and 570–580 cm^−1^, which was ascribed to the respective molecular stretch and bending of hydroxyl groups (OH) on magnetite SPIONs surface and Fe-O bonds, respectively [37,38]. The two sharp peaks observed for pristine CLA (Figure 2b) at around 2922 cm^−1^ and 2852 cm^−1^ belonged to the ν_sym_ and ν_asym_ CH_2_ group stretching [39]. Another sharp peak at around 1710 cm^−1^ was indicative of νC=O of the carboxyl group, and that at around 1408 cm^−1^ to the CH_3_ group bending [39,58]. There was no significant distinction in the spectrum of the CLA-coated SPIONs (Figure 2c) and those of pristine CLA and SPIONs, except that the band at 3420 cm^−1^ initially present in the absorption spectrum of the uncoated SPIONs attributed to surface OH groups disappeared with the CLA-coated SPIONs, due to chemisorption of CLA to the surface of the SPIONs (Figure 2f). Characteristic peaks of PTX (Figure 2d) were observed at around 3500, 3479, 2976–2885, 1731, and 1244 cm^−1^, assigned to the OH, N-H, and CH_2_ vibrations, C=O, and ester groups, respectively [40,59]. Similar peaks were present in the spectrum of the CLA-coated PTX-SPIONs (Figure 2e), indicating that PTX was successfully adsorbed onto the surface of the CLA-coated SPIONs and the partitioning did not alter the chemical structure of PTX.

### 3.3. Quantification of CLA Content on CLA-Coated SPIONs

The amount of CLA coated onto SPIONs was ascertained by TGA. Figure 3 presents TGA data showing % weight loss as function of temperature, for pristine SPIONs (Figure 3a) and CLA-coated SPIONs (Figure 3b). Pristine SPIONs demonstrated an initial weight loss at below 200 °C, which is attributed to the evaporation of adsorbed water on the surface [37,38]. The total weight loss of 5.7% was thus recorded for pristine SPIONs. Meanwhile, CLA-coated SPIONs exhibited a two-step weight loss pattern. The initial weight loss was observed at below 200 °C, complementary to pristine SPIONs (water desorption), and the second weight loss was observed between 200–450 °C, which is attributed to the removal of CLA layer from SPIONs surface. The content of CLA on SPIONs could therefore be estimated from 200–450 °C and was found to be 10.3%. Previous studies have reported fatty acid contents in the range of 10–14%, when fatty acids such as linoleic acid and palmitic acid were coated onto SPIONs and analyzed by TGA [38,40]. The TGA results correlate with VSM data, which demonstrated a slight reduction in saturation magnetization of CLA-coated PTX-SPIONs (60 emu/g) compared to pristine SPIONs (65 emu/g) due to the presence of CLA on magnetite.

### 3.4. Determination of the Structural Crystallinity of the CLA-Coated PTX-SPIONs

The diffraction patterns presented in Figure 4 show the crystalline nature of the SPIONs before (Figure 4a) and after CLA coating (Figure 4b) and PTX loading (Figure 4c) (CLA-coated PTX-SPIONs). The obtained XRD patterns show similar diffraction peaks (2θ) at 30.5°, 35.8°, 43.5°, 57.3°, and 63.1°, for all samples, characteristic of magnetite [37,38], indicating that pristine SPIONs and the CLA-coated PTX-SPIONs were purely magnetite (Fe_3_O_4_) and crystalline in nature. This correlates with the Raman spectroscopy analysis and further ascertains that CLA did not affect the crystallinity of the SPIONs. These results agree well with existing studies reported by Sawisai et al. (2019), for SPIONs coated with linoleic acid [38]. The difference in peak intensity between SPIONs and the CLA-coated SPIONs was attributed to the CLA coating process that reduced the peak intensity. Meanwhile, the diffractogram for CLA-coated PTX-SPIONs showed additional diffraction peaks (2θ) at 22.4° and 25.1° corresponding to PTX crystallinity [60,61]. This indicated the presence of PTX in the formulation in its crystalline form.

### 3.5. Evaluation of the Iron Oxide Core Integrity of the SPIONs

Figure 5 presents Raman spectra of (a) pristine SPIONs and (b) the CLA-coated PTX-SPIONs. Both spectra show a single prominent band at 670 cm^−1^ representing magnetite (Fe_3_O_4_) origin, attributed to the A_1g_ photon mode [38]. Similar observation was made for CLA-coated SPIONs (data not shown). The single peak observed provides evidence that the synthesized SPIONs were purely magnetite, and there was no phase transition after the CLA coating process. Previously published literature has reported that the phase transition of magnetite to maghemite (γ-Fe_2_O_3_) mainly occurs at higher temperatures (200–600 °C) [62].

### 3.6. Assessment of the Magnetic Properties of the CLA-Coated PTX-SPIONs

The magnetization analysis was conducted to further confirm the superparamagnetism of the synthesized SPIONs prior and post modification with the CLA coating and the adsorption of PTX (CLA-coated PTX-SPIONs). Shown in Figure 6a,b are pristine SPIONs and CLA-coated PTX-SPIONs localized by a magnet, indicative of inherent magnetism. Magnetization hysteresis loops for pristine SPIONs and the CLA-coated PTX-SPIONs are depicted in Figure 6c,d, with corresponding saturation magnetization and coercivity values presented in Table 1. Both samples were confirmed to be superparamagnetic as evidenced by the S-type behavior and nearly no remanence in the hysteresis. However, CLA-coated PTX-SPIONs exhibited a slightly reduced magnetic saturation compared to the pristine SPIONs, which is attributed to the CLA coating process resulting in slight reduction of exposed Fe_3_O_4_ content and reduced effect of the magnetic field. Moreover, the coercivity values of the samples varied, but were both <100 Oe, thus further demonstrating superparamagnetism and suitability for biomedical application [63,64]. In essence, the CLA-coated PTX-SPIONs maintained the superparamagnetic behavior and met the criteria for biomedical applications.

### 3.7. Assessment of the PTX Adsorption Efficiency and In Vitro Release from the CLA-Coated PTX-SPIONs

PTX loading onto the SPIONs is a crucial step in their application as an anticancer nanomedicine for potential subcutaneous injection. Essentially, the drug should be loaded in a way that does not compromise its functionality, while ensuring adequate release from the nanosystem upon reaching the target site [19]. In the present study, PTX was successfully loaded onto the CLA-coated SPIONs, achieving an Adsorption Efficiency (%AE) of 98.5% and %DLC of 8.9%. Conventionally, it is challenging to encapsulate hydrophobic drugs and to achieve controlled drug release [65]. However, in the present study, PTX was preferentially adsorbed (98.5%) onto the CLA coating due to the hydrophobicity of CLA, which facilitated adsorption and sustained release over time from the SPIONs architype.

This efficient drug loading achieved validates the strategy of CLA-coating process over classical conjugation for hydrophobic drugs such as PTX. Dilnawaz et al., reportedly achieved ~95% loading efficiency for PTX and rapamycin in glycerol monooleate (GMO)-coated SPIONs, where drugs were co-encapsulated in the hydrophobic GMO crust [40].

The in vitro drug release study was conducted to quantify the amount of PTX released from the CLA-coated SPIONs at pH 7.4 and 6.8, representative of physiological pH and a tumor microenvironment (TME), respectively [42]. The PTX release over 24 h is presented in Figure 7. Evidently, the release of PTX was relatively higher at pH 6.8 than pH 7.4. A maximum cumulative release of 94% was obtained over 12 h at pH 6.8, indicating sufficient sustained release of PTX from the CLA coating, which is attributed to the disruption of the CLA-PTX hydrophobic interaction at a relatively acidic pH of 6.8 over time, resulting in PTX release. A similar observation was reported with PTX-loaded SPIONs coated with GMO [40,66]. The cumulative drug release at pH 7.4 was <18%, implying that the CLA-coated PTX-SPIONs has the potential to control the release of PTX to lung cancer tumors (as indicated by the higher PTX release at pH 6.8) while having minimal cytotoxicity to healthy cells (evidenced by lower drug release at physiological pH of 7.4). Furthermore, shielding of PTX at pH 7.4 shows the potential of the nanosystem to enhance the bioavailability of PTX.

### 3.8. Evaluation of the Anti-Proliferative Activity Using a Cell Viability Study on the CLA-Coated PTX-SPIONs

The MTT assay was used to evaluate the cytotoxicity of the CLA-coated PTX-SPIONs in a lung adenocarcinoma cell line (A549) by measuring cell viability after 72 h of treatment. Cells were treated with varying concentrations (3.215–100 µg/mL) of pristine SPIONs, CLA, and CLA-coated PTX-SPIONs, while pristine PTX was used as a positive control (Figure 8). The % cell viability for cells treated with pristine SPIONs was comparatively higher after 72 h, with 95.5% and 91.3% obtained at the lowest and highest treatment concentration, respectively. This indicated that the pristine SPIONs did not have significant cytotoxicity on A549 cells.

This further validated the use of SPIONs as bio-imaging agents in cancer diagnosis, since they do not confer any significant harm to cells. Generally, SPIONs are applied in magnetic resonance imaging (MRI) for bio-imaging because of their excellent image contrasting properties attributed to their inherent magnetism [67]. The anti-proliferative activity of pristine *trans*-10, *cis*-12 CLA was observed, with recorded 69.8% cell viability of A549 cells at highest treatment concentration (100 µg/mL) after 72 h. This showed that *trans*-10, *cis*-12 CLA resulted in 30.2% inhibition of lung adenocarcinoma cells, and adds to previous works which have demonstrated that *trans*-10, *cis*-12 CLA suppresses cancer proliferation [32,33,68].

Treatment with the CLA-coated PTX-SPIONs highly suppressed A549 cell proliferation after 72 h in a dose-dependent manner (Figure 8a). Treatment at highest concentration (100 µg/mL) resulted in only 17.1% cell viability and treatment with the lowest concentration (3.125 µg/mL) resulted in 24.2% viability. Interestingly, when compared to the pristine PTX (Figure 8b), treatment with lower concentrations of CLA-coated PTX-SPIONs of 3.125 and 6.25 µg/mL yielded a % cell viability of 24.2% and 21.6%, respectively. This demonstrated the more enhanced anti-proliferative activity than recorded with the highest concentration of pristine PTX (10 µg/mL), which resulted in a cell viability value of 25.3%. Essentially, this showed that a smaller dose of the CLA-coated PTX-SPIONs would be required to elicit equivalent anti-proliferative activity as exerted by a ~3-fold dose of pristine PTX. The results then suggest that the CLA-coated PTX-SPIONs enhance the efficacy of PTX in suppressing A549 cell proliferation.

The IC_50_ for PTX, and the CLA-coated PTX-SPIONs were measured to be 0.7 and 0.3 µg/mL, respectively. This substantiates the comparatively higher anti-proliferative activity of the CLA-coated PTX-SPIONs as evidenced by the lower % cell viability compared to pristine PTX. An IC_50_ of 0.9 µg/mL for pristine PTX in A549 cells has been reported previously by Jiang et al. [69], thus demonstrating that pristine PTX employed in this study had comparable cytotoxicity.

The enhanced anti-proliferative activity of the CLA-coated PTX-SPIONs observed is attributed to the presence of the CLA coating onto the SPIONs. The two biologically active CLA isomers (*trans*-10, *cis*-12 and *cis*-9, *trans*-11) have been reported previously to show anticancer activity with studies suggesting that in combination they suppress tumor growth via DNA synthesis inhibition and lipid peroxidation modulation, amongst other mechanisms [28,31,70,71]. As such, it is presumed that, in addition to acting as a surface coating agent that allowed for maximal partitioning of PTX onto the SPIONs, the *trans*-10, *cis*-12 CLA isomer imparted additional anticancer action in tandem with the PTX on the A549 cells, thus resulting in its increased anti-proliferative activity. Moreover, the sustained release of PTX observed from the CLA-coated PTX-SPIONs (Figure 7) is linked to the outcome of the cell viability study, demonstrating that the prolonged release of PTX over time resulted in enhanced cytotoxicity. Thus, it can be deduced that the CLA-coated PTX-SPIONs offer superior anti-proliferative activity against A549 cancer cells over pristine PTX.

## 4. Conclusions

SPIONs, which are purely magnetite (Fe_3_O_4_), were successfully synthesized, coated with *trans*-10, *cis*-12 CLA, and loaded with PTX. The superparamagnetic behavior of the CLA-coated PTX-SPIONs was maintained and verified by VSM analysis with a mean particle size of <100 nm. The surface coating of SPIONs with CLA facilitated enhanced and efficient PTX adsorption (98.5%) via hydrophobic interactions. PTX was partitioned onto the CLA surrounding the SPIONs as confirmed by physicochemical characterization. Sustained release of PTX was achieved up to 94% in a simulated TME at pH 6.8 after 24 h with a significantly reduced release at pH 7.4 (physiological milieu). A549 cells treated with the CLA-coated PTX-SPIONs showed a comparatively lower viability compared to cells treated with pristine PTX, confirming the potential enhancement of the anticancer activity of PTX in suppressing lung cancer proliferation (potentially in NSCLC) using the synthesized superparamagnetic nanosystem. Results from this study also validate the proof-of-concept of using CLA as a coating agent onto SPIONs to enhance the activity of hydrophobic anticancer drugs with future functionalization that could provide more active in vivo targeting of NSCLC as a nanomedicine.

## Figures and Tables

**Figure 1 pharmaceutics-14-00829-f001:**
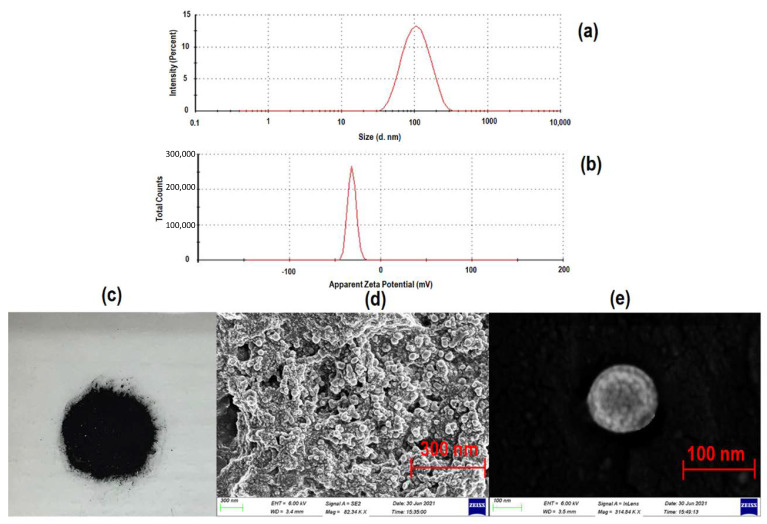
A visualization of the morphological attributes of the CLA-coated PTX-SPIONs, showing (**a**) average hydrodynamic particle size of 96.5 ± 0.6 nm (PDI = 0.1 ± 0.02), (**b**) zeta potential of −27.3 ± 1.9 mV, and (**c**) digital image of a lyophilized sample with corresponding SEM images showing overall morphology (**d**) at 300 nm scale (82.34 KX magnification) and (**e**) 100 nm scale (314.84 KX) (isolated particle), showing a particle size of <100 nm.

**Figure 2 pharmaceutics-14-00829-f002:**
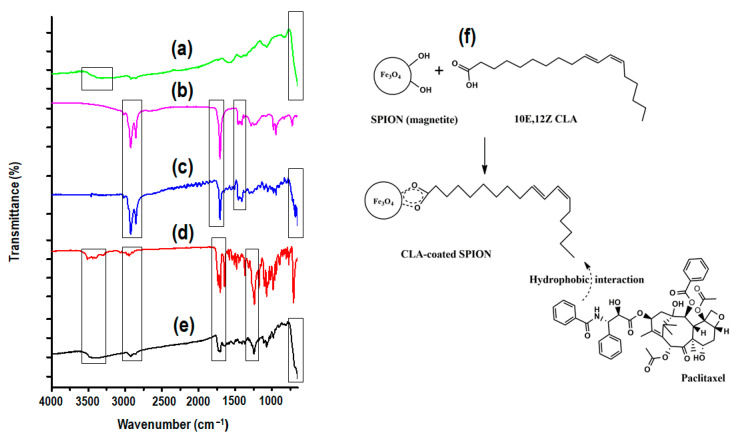
FT-IR spectra of (**a**) pristine SPIONs, (**b**) pristine CLA, (**c**) CLA-coated SPIONs, (**d**) pristine PTX, (**e**) CLA-coated PTX-SPIONs, and (**f**) schematic representation of CLA chemisorption onto SPIONs and CLA-PTX hydrophobic interaction.

**Figure 3 pharmaceutics-14-00829-f003:**
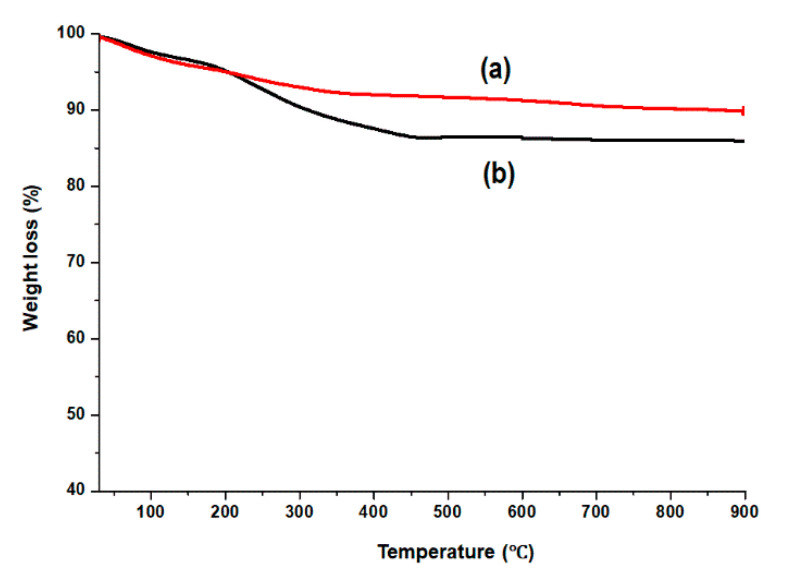
TGA thermograms of (**a**) pristine SPIONs and (**b**) CLA-coated SPIONs demonstrating weight loss (%) over a temperature range of 30–900 °C.

**Figure 4 pharmaceutics-14-00829-f004:**
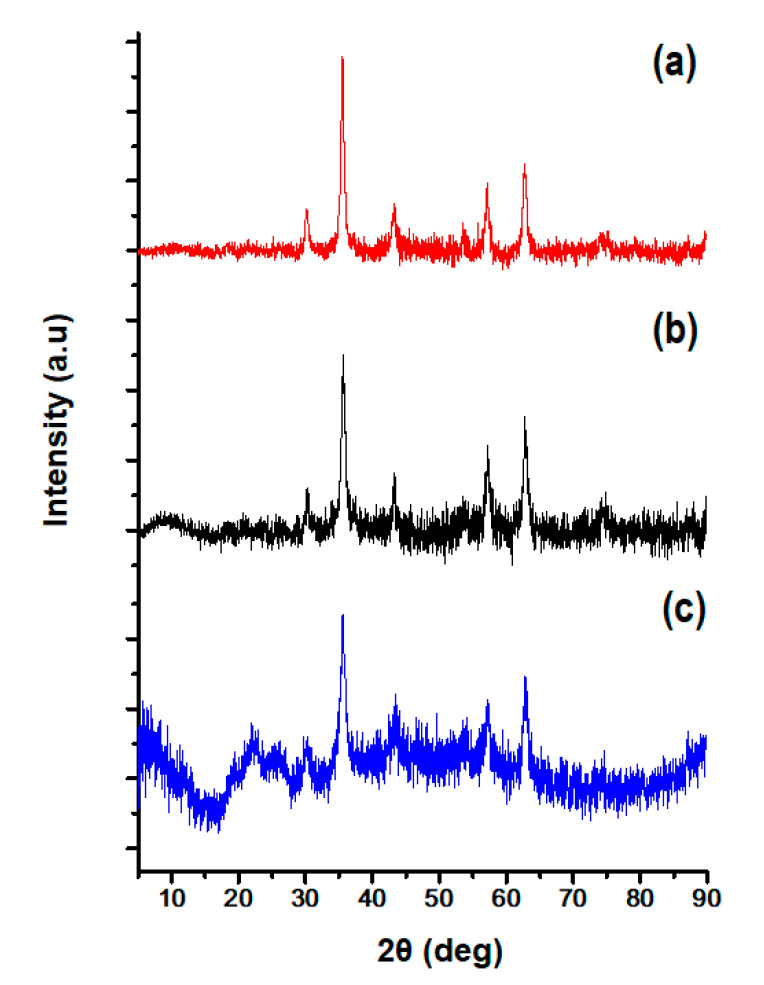
XRD spectra of (**a**) pristine SPIONs, (**b**) CLA-coated SPIONs, and (**c**) CLA-coated PTX-SPIONs showing the crystalline nature of the SPIONs before and after the CLA coating and PTX loading.

**Figure 5 pharmaceutics-14-00829-f005:**
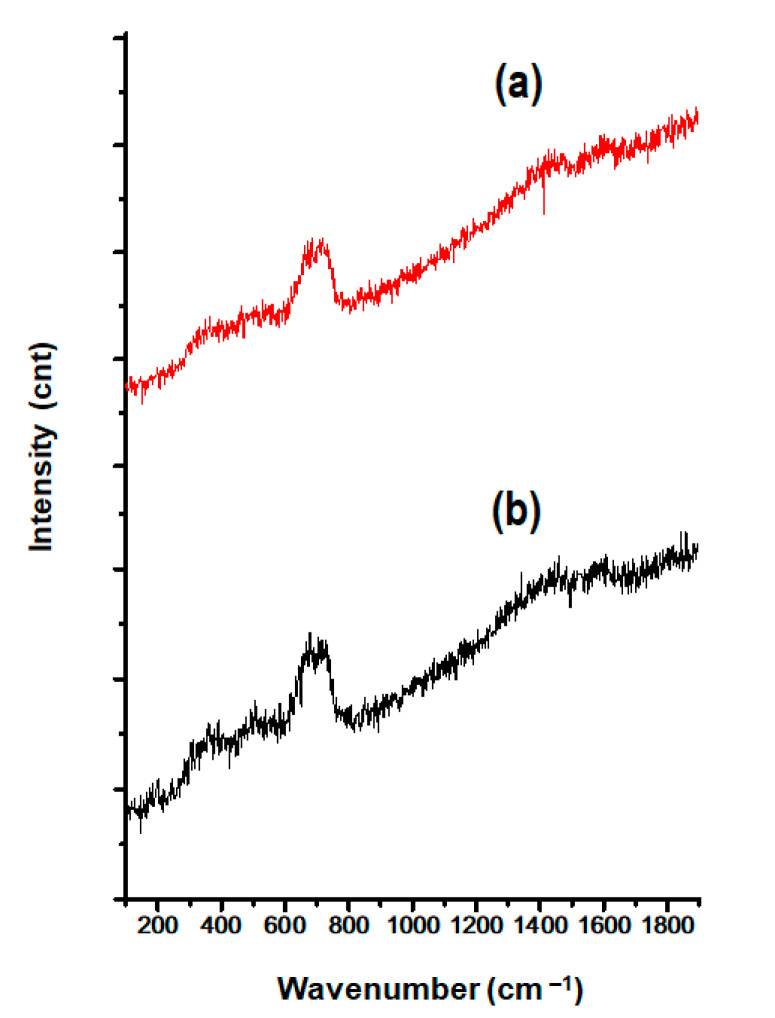
Raman spectra of (**a**) pristine SPIONs and (**b**) the CLA-coated PTX-SPIONs depicting the presence and integrity of the magnetite.

**Figure 6 pharmaceutics-14-00829-f006:**
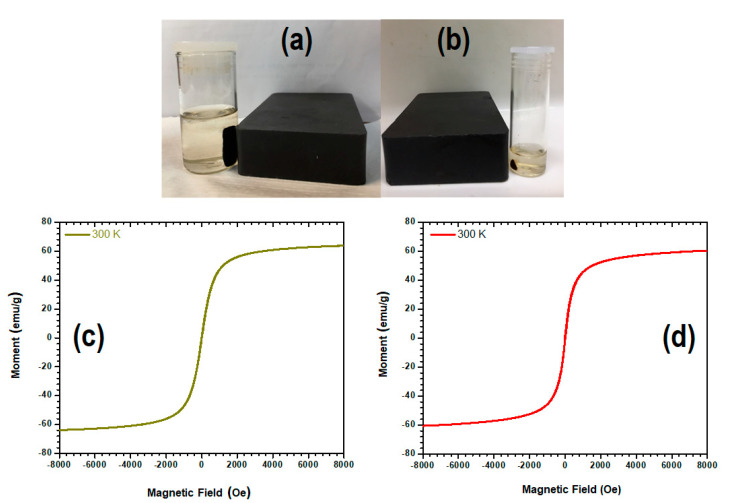
A depiction of (**a**) pristine SPIONs and (**b**) CLA-coated PTX-SPIONs localized by a magnet, and magnetization hysteresis loops of (**c**) pristine SPIONs and (**d**) CLA-coated PTX-SPIONs at 300 K and magnetic field range of −8000 to 8000 Oe.

**Figure 7 pharmaceutics-14-00829-f007:**
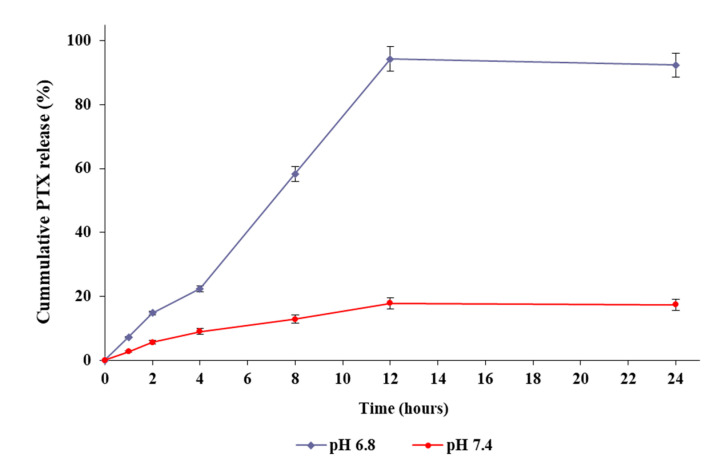
In vitro PTX release profiles from the CLA-coated PTX-SPIONs at pH 6.8 and pH 7.4. Data from experiments are presented as the mean ± SD, *n* = 3.

**Figure 8 pharmaceutics-14-00829-f008:**
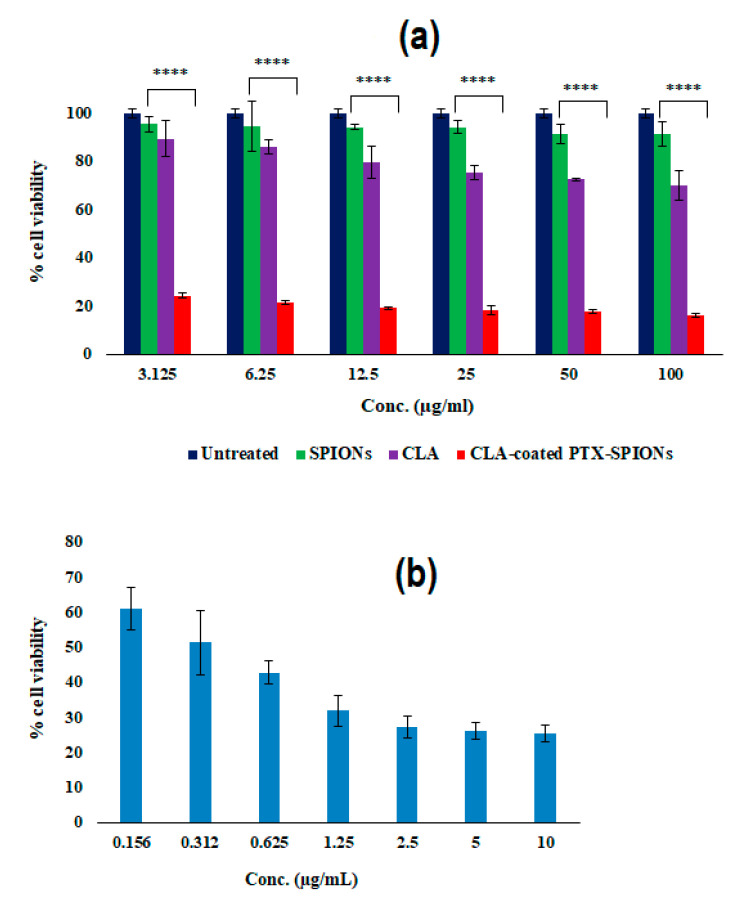
Profiles showing % cell viability of lung adenocarcinoma cells (A549) treated with varying concentrations of (**a**) pristine SPIONs (negative control) and the CLA-coated PTX-SPIONs, and (**b**) pristine PTX (positive control) after 72 h. Data presented as mean ± SD, *n* = 3 (**** denote *p* < 0.0001, when CLA-coated PTX-SPIONs are compared to SPIONs and CLA at the same concentrations).

**Table 1 pharmaceutics-14-00829-t001:** Saturation magnetization and coercivity of pristine SPIONs and the CLA-coated PTX-SPIONs as determined by VSM analysis.

Sample	Saturation Magnetization (emu/g)	Coercivity (Oe)
Pristine SPIONs	65	50
CLA-coated PTX-SPIONs	60	29

## Data Availability

The data presented in this study is available on request from the corresponding author.

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
