# Peer review of "Synthesis of Novel Conjugated Linoleic Acid (CLA)-Coated Superparamagnetic Iron Oxide Nanoparticles (SPIONs) for the Delivery of Paclitaxel with Enhanced In Vitro Anti-Proliferative Activity on A549 Lung Cancer Cells"

_pharmaceutics, 2022, doi:10.3390/pharmaceutics14040829_

Round 1

Reviewer 1 Report

The standard deviation value in the particle size value is too small. Check the value in the sample analysis file, and enter the correct standard deviation. This remark must be taken into account for all the results described in the article.

Graphs with the results of experiments on cells are not clear replace them with better quality drawings.

2.1. Materials

Specify the solvents used, need to add - magnetite standard, PBS, Tween, dialysis membrane.

2.2.1. Synthesis of the SPIONs

Add a reference to the method you modified.

«The formed SPIONs were centrifuged at 5000 rpm (Eppendorf Centrifuge 5804, Hamburg, Germany) for 30 minutes to eliminate any unreacted material.»

Specify the temperature at which centrifugation was performed.

2.2.2. Coating of the SPIONs with trans-10 cis-12 (10E, 12Z) conjugated linoleic acid
«A 40 µL from a 100 mg/mL 10E, 12Z CLA solution»

In what solvent was the solution prepared? Please add clarifications to the text.

«was added drop-wise into a 1.5 mL suspension of SPIONs (20 mg in ethanol)» - It is necessary to add the volume of ethanol or the concentration of the suspension.

What method was used to determine the amount of conjugated linoleic acid? Add this part.

2.2.3. Self-assembled loading of PTX into 10E, 12Z CLA-SPIONs
Why was methanol used in this method when ethanol was used before?

Add a reference to this method if it has already been described.

Specify the temperature during centrifugation.

Were the centrifugation conditions of CLA-coated PTX-SPION adjusted to separate from free PTX? Usually, under these conditions, free PTC settles together with nanoparticles.

«The supernatant (with free drug) was collected for quantifying the PTX loading capacity and adsorption efficiency via UV spectrophotometry (Cary™ 50, Varian Inc., USA). The PTX-loaded CLA-coated SPIONs were washed thrice with distilled water and subsequently lyophilized (FreeZone-50C Benchtop Freeze Dryer, Labconco, USA) to attain free-flowing powdered CLA-coated PTX-SPIONs.»

Why is only the first supernatant analyzed? Subsequent washes can also desorb PTX, and all supernatants must be counted to analyze the remaining PTX in CLA-coated PTX-SPIONs because you are not analyzing directly, but rather by subtracting the value in the supernatant from the amount added.

2.3.2. Determination of particle size, zeta potential, and morphology

«surface charge of the synthesized SPIONs» is not determined by Dynamic light scattering measurements. Specify the correct method for measuring the zeta potential. It is necessary to add the protocol according to which the measurements were taken.

2.3.4. Iron oxide core mapping of the SPIONs

«Raman spectroscopy (SENTERRA II, BRUKER, Germany) was employed because it is a precise method to characterize the magnetite nature of the iron oxide core of the SPI-ONs before and after the CLA-coating enhancement process.»

Please cite a source in the literature for this statement.

2.3.6. Determination of PTX-loading capacity and adsorption efficiency within the CLA-coated SPIONs

Similar remarks about the centrifugation mode for separation from free PTX. Correct in accordance with the changes in paragraph 2.2.3.

Specify the temperature at which centrifugation was performed.

What was PTX dissolved in to construct a standard calibration curve?

You must specify the dimension in formulas 1 and 2 for "?????? ?? ???? ?????d", "?????? ?? ???????? ?? ??????????t", "????? ????ℎ? ??????????n".

2.3.7. Evaluation of in vitro release of PTX from the CLA-coated SPIONs

Give references to this method in the descriptions for the analysis of such systems. If there is no such source, explain why you took this amount of tween.

Add information about the total volume in which the samples were dissolved and incubated during the experiment.

Have you made any new PTX calibration plots in the release environment for subsequent PTX quantification?

2.4.1. Cell viability study using a MTT assay

Why single out paragraph «2.4.1. Cell viability study using a MTT assay», if not 2.4.2.? Remove unnecessary numbering.

«There-after, 100 µL of the solubilizing agent was added to dissolve formed formazan crystals, followed by overnight incubation at 37℃ and 5% CO2

What was used as a solubilizing agent? It is necessary to supplement the methodology and add materials to the section.

«100 µL of the solubilizing agent was added to dissolve formed formazan crystals, followed by overnight incubation at 37℃ and 5% CO2»

Why was incubation carried out overnight? The crystals dissolve within 30 minutes. After a long incubation, the color of the solution changes, which affects the measured values. Provide a link to the methodology and rationale for why the non-classical version of the MTT analysis was chosen.

Equation 3.

Write a transcript of all the components of the formula, indicating the units of measurement.

  1. Results and Discussion
    Sizing and zeta potential results of CLA-coated SPIONs should be added.

258: «The synthesized SPIONs were therefore in a desirable range (10-100 nm) for potential targeted PTX delivery in NSCLC.»

Judging by Figure 1a, the range is from 30 to 350 nm. Change the description in the text with the correct wording.

275-276: «the CLA-coated PTX-SPIONs were spherical in shape and having a relatively rugged surface (indicative of adsorbed PTX onto the CLA coating)»
There are no clear results for this statement. Need to rewrite with comments.

Figure 1. с.

The results shown do not look like a SEM micrograph, it looks more like a TEM. Check the correctness of the specified method. If the micrograph was obtained by TEM, it is necessary to complete the technique. There is no scale in the figure, it needs to be added.

Figure 1. d, e.

The captions under the figure have their own scale, which differs in size from the scale given by the authors on the microphotographs themselves.

Figure 3.

You need to add a curve for CLA-coated PTX-SPIONs.

Figure 4.

Need to add a curve for CLA-coated SPIONs.

3.7. Evaluation of the anti-proliferative activity using a cell viability study on the CLA-coated PTX-SPIONs

Add the results for CLA-coated SPIONs, CLA to the figure and describe them in the text.

What were the samples dissolved in for analysis? Add relevant information to paragraph 2.4.1.

There is no statistical analysis of the obtained results. Analyze the presented results, add the appropriate section to the materials and methods.

All samples must be submitted within the same concentration range. It is necessary to combine all the results into one graph.

Reviewer 2 Report

SIGNIFICANCE OF THE WORK:

In this manuscript, it is described the preparation of a drug delivery system based on SPIONs coated with conjugated linoleic acid (CLA) and paclitaxel (PTX). Its potential application in targeted therapy in NSCLC is discussed.

METHODOLOGY:

The prepared materials have been thoroughly characterized by typical methods.

TEXT:

The manuscript is well-written, and the figures are well crafted

NOVELTY

To the best of my knowledge the system reported is new.

COMMENTS:

Although the design and synthesis of the system is quite interesting, some points should be addressed:

  • According to Figure 1 a) the size of CLA-coated PTX-SPIONs nanoparticles ranges from 30 to 300 nm. Could the authors explain such polydispersity?

  • TEM images of more than one nanoparticles should be shown.

  • In line 445, the authors attribute the enhanced antiproliferative activity of the nanoparticles to the presence of CLA (Linoleic acid) (“the enhanced anti-proliferative activity of the CLA-coated PTX-SPIONs observed is attributed to the presence of the CLA coating onto the SPIONs”). To demonstrate this hypothesis, combined treatments with CLA and PXT must be conducted. Furthermore, release of CLA from the particles must be determined. These experiments should allow determining the combinatorial index of the drugs.

Reviewer 3 Report

I have read the manuscript “Synthesis of Novel Conjugated Linoleic Acid (CLA)-Coated Superparamagnetic Iron Oxide Nanoparticles (SPIONs) for the Delivery of Paclitaxel with Enhanced In Vitro Anti-Proliferative Activity on A549 Lung Cancer Cells” by Lindokuhle M. Ngema et al. (MS # pharmaceutics-1659580) submitted for the publication in Pharmaceutics.

The authors reported their investigations on the synthesis of SPIONs and their functionalization with linoleic acid for the delivery of PTX. After the chemical-physical characterization nanoparticles were successfully tested against A549 cancer cells.

The topic is interesting and timely, and the manuscript deserves publication in Pharmaceutics after the following minor revisions:

  1. Line 112: the sentence is misunderstanding as one used pure CLA-coated SPIONs;
  2. Line 276: a SEM image cannot indicative of adsorption;
  3. Figure 1.a and caption: Peak of the size in intensity looks like larger than 100nm;
  4. Figure 1.e: picture quality is too low to reveal the rugged surface;
  5. Figure 2. X-axis legend needs an apex (cm-1);
  6. Line 319 and Figure 3: diffractograms show some peak shifts. What about?
  7. Figures 5.a and 5.b: please, use the same magnification and angle of view;
  8. Figures 5.c and 5.d: please, use the same vertical scale for a better comparison;
  9. Line 380: In order to confirm the successful loading of PTX even with a %DLC = 8.9%, please, compare with similar data present in literature for PTX drug delivery systems;
  10. Figure 6: what about the decrease in PTX release at a pH value of 6.8 after 24 h?
  11. Figures 7.a and 7.b: please, use points rather than commas for decimals;
  12. Figure 7.a; %cell viability do not seem to be much sensitive to CLA-coated PTX-Spions concentrations for values larger than 12.5 ug/ml. What about? What happens for concentration lower than 3.125 ug/ml?

Reviewer 4 Report

The article titled “Synthesis of Novel Conjugated Linoleic Acid (CLA)-Coated Superparamagnetic Iron Oxide Nanoparticles (SPIONs) for the Delivery of Paclitaxel with Enhanced In Vitro Anti-Proliferative Activity on A549 Lung Cancer Cells” by Ngema et al. has been reviewed where the authors have prepared and presented CLA-coated SPIONS as the novel nanoparticles for Paclitaxel delivery and investigated their anti-proliferative activity on lung cancer cells. The work has been designed well and sufficient data has been provided. However, in my opinion there are some concerns to be addressed before the work can be accepted for publication.

  1. There are various works published where SPIONS are used for drug delivery not only for cancer treatment, but also for various other purposes. Thus, the novelty of this work is not so high and hence in my opinion the word “Novel” is not suitable in the title of this work. Using the idea of CLA coating does not bring much novelty to this either.
  2. The CLA-coated SPIONs are claimed to show enhanced therapeutic activity in NSCLC therapy, however it is not clear how the authors have drawn such conclusion that these particles have shown enhanced activity? Is it compared to some other particles or systems, not clear!
  3. The authors have designed CLA-coated and PXT-loaded SPIONS. Will it be possible to present schematically the interactions responsible for both the coating and loading process? Also, providing a brief explanation on how these interactions favored the coating and loading would make this more clearly understandable.
  4. For Figure 1a, mentioning the PDI value alongside the particle size would be informative.
  5. What is the solvent used for the zeta potential measurements, water or any buffer? It is known that generally the pH of pure water can vary between 5 and 7, hence the role of solvent is important here.
  6. In the X-axis of Figure 7, should the ‘commas’ in 3,125; 6,25; 12,5 not be actually ‘decimal points’? This needs rectification.
  7. The authors have not mentioned anything about the clearance of the SPIONs from a physiological body after the delivery of PTX. Will these not be toxic to healthy cells? Can the authors provide some insight on their understanding about the clearance these nanoparticles from body?

Round 2

Reviewer 1 Report

Thanks for the edits.

Reviewer 2 Report

Thank you very much for the revised manuscript. The authors have answer my questions and comments.

Reviewer 4 Report

The authors have significantly improved the manuscript by revising it. Hence, I recommend it for publication.